# Adalimumab and ABP 501 in the Treatment of a Large Cohort of Patients with Inflammatory Arthritis: A Real Life Retrospective Analysis

**DOI:** 10.3390/jpm12030335

**Published:** 2022-02-23

**Authors:** Andrea Becciolini, Simone Parisi, Rosalba Caccavale, Elena Bravi, Federica Lumetti, Romina Andracco, Alessandro Volpe, Lucia Gardelli, Francesco Girelli, Eleonora Di Donato, Daniele Santilli, Gianluca Lucchini, Maria Chiara Ditto, Ilaria Platè, Eugenio Arrigoni, Flavio Mozzani, Michele Riva, Antonio Marchetta, Enrico Fusaro, Gilda Sandri, Carlo Salvarani, Marino Paroli, Alarico Ariani

**Affiliations:** 1Rheumatology Unit, Department of Medicine and Internal Medicine, University Hospital of Parma, 43121 Parma, Italy; eleonoradidonato@ymail.com (E.D.D.); dsantilli@ao.pr.it (D.S.); gianluca.lucchini76@gmail.com (G.L.); fmozzani@ao.pr.it (F.M.); mriva@ao.pr.it (M.R.); atendoro@gmail.com (A.A.); 2Rheumatology Unit, Department of General and Specialistic Medicine, Azienda Ospedaliero-Universitaria, Città della Salute e della Scienza di Torino, 10126 Turin, Italy; simone.parisi@hotmail.it (S.P.); mariachiaraditto@gmail.com (M.C.D.); fusaro.reumatorino@gmail.com (E.F.); 3Department of Medical-Surgical Sciences and Biotechnologies, Sapienza University of Rome, Polo Pontino, 04100 Latina, Italy; rosalba_caccavale@yahoo.it (R.C.); marino.paroli@uniroma1.it (M.P.); 4Department of Rheumatology, Ospedale Guglielmo da Saliceto, 29121 Piacenza, Italy; bravielena@yahoo.it (E.B.); ilariaplate@gmail.com (I.P.); e.arrigoni@ausl.pc.it (E.A.); 5Rheumatology Unit, Azienda USL of Modena and AOU Policlinico of Modena, 41100 Modena, Italy; fedelumetti@gmail.com; 6Distretto Socio Sanitario ASL 1 Imperiese, 18001 Imperia, Italy; r.andracco@libero.it; 7Rheumatology Unit, IRCCS Sacro Cuore Don Calabria, 37024 Negrar, Italy; alessandro.volpe@sacrocuore.it (A.V.); antonio.marchetta@sacrocuore.it (A.M.); 8Internal Medicine Unit, GB Morgagni Hospital, 47121 Forli, Italy; lucia.gardelli@auslromagna.it (L.G.); fgirodoc@gmail.com (F.G.); 9Rheumatology Unit, University of Modena and Reggio Emilia, 41100 Modena, Italy; gilda.sandri@unimore.it (G.S.); carlo.salvarani@unimore.it (C.S.)

**Keywords:** rheumatoid arthritis, psoriatic arthritis, axial spondylarthritis, adalimumab, ABP 501

## Abstract

The recent introduction of ABP 501, an adalimumab biosimilar, in the treatment of rheumatic diseases was supported by a comprehensive comparability exercise with its originator. On the other hand, observational studies comparing adalimumab and ABP 501 in inflammatory arthritis are still lacking. The main aim of this study is to compare the clinical outcomes of the treatment with adalimumab, both the originator and ABP 501, in a large cohort of patients affected by autoimmune arthritis in a real life setting. We retrospectively analysed the baseline characteristics and the retention rate in a cohort of patients who received at least a course of adalimumab (originator or ABP 501) from January 2003 to December 2020. We stratified the study population according to adalimumab use: naive to original (oADA), naive to ABP 501 (bADA) and switched from original to ABP 501 (sADA). The oADA, bADA and sADA groups included, respectively, 724, 129 and 193 patients. In each group, the majority of patients had a diagnosis of rheumatoid arthritis. The total observation period was 9805.6 patient-months. The 18-month retentions rate in oADA, bADA and sADA was, respectively, 81.5%, 84.0% and 88.0% (*p* > 0.05). The factors influencing the adalimumab retention rate were an axial spondylarthritis diagnosis (Hazard Ratio (HR) 0.70; *p* = 0.04), switch from oADA to ABP 501 (HR 0.53; *p* = 0.02) and year of prescription (HR 1.04; *p* = 0.04). In this retrospective study, patients naive to the adalimumab originator and its biosimilar ABP 501 showed the same retention rate. Patients switching from the originator to biosimilar had a higher retention rate, even though not statistically significant, when compared to naive.

## 1. Introduction

The treatment of inflammatory arthritides, such as rheumatoid arthritis (RA), axial spondylarthritis (axSpA) and psoriatic arthritis (PsA), has deeply changed in the last two decades due to the introduction of biological disease-modifying antirheumatic drugs (bDMARDs). The first bDMARDs utilized in clinical practice for the treatment of RA, axSpA and PsA were the tumour necrosis factor alpha inhibitors (TNFi) [1]. Randomised controlled trials and observational real-world studies highlighted that TNFi use in inflammatory arthritides was able to control disease activity, improve patients’ quality of life and reduce damage progression [2]. In recent years, as the originator bDMARDs’ patents expired, less expensive biosimilar drugs have emerged. To date, biosimilars of three TNFi (infliximab, etanercept and adalimumab (ADA)) have been approved in rheumatology by European Medicines Agency (EMA). ABP 501, an ADA biosimilar, was approved in March 2017 for the same reference product rheumatological indication. ABP 501 was demonstrated to be similar in structure, function and pharmacokinetics to its originator in preclinical and phase I clinical trials [3,4,5]. Furthermore, a phase III equivalence study was carried out in 526 RA patients highlighting a comparable efficacy and safety [6]. Finally, the subsequent open-label extension study included 467 RA patients and demonstrated the long-term efficacy and safety of ABP 501 even in those previously treated with the reference product [7].

Despite the evidence from registration studies, the external validity of randomised controlled trials may be significantly hindered by stringent inclusion and exclusion criteria, thus limiting its generalisability to real-world clinical practice [8]. In this perspective, the implementation of biosimilars in real-life practice is still a subject of controversy among rheumatologists. Therefore, large real-life observational studies are warranted to evaluate the effectiveness of ABP 501. In observational registries, the retention rate is considered a reliable indicator of treatment effectiveness since it is committed to drug efficacy and safety. To date, only few real-world data have been published concerning the use of ABP 501, mainly in the context of inflammatory bowel diseases and psoriasis [9,10,11,12,13]. With the exclusion of these reports, no other data on ABP 501 use in both reference product-naive and experienced inflammatory arthritides patients are available. The main aim of this large multicentre observational retrospective study is to verify if the adalimumab retention rate is similar in RA, axSpA and PsA naive patients (both to original and ABP 501 biosimilar) and in those who switched from original to ABP 501 biosimilar.

## 2. Materials and Methods

The BIRRA (BIologics Retention Rate Assessment) observational retrospective study was carried out following the Declaration of Helsinki principles and approved by the local Ethics Committee (Comitato Etico dell’Area Vasta Emilia Nord, protocol code 34713, approved on 28 August 2019).

### 2.1. Patients

We included patients with a clinical diagnosis of RA, PsA and axSpA from eight rheumatology units in Italy. All patients received ADA, both originator or biosimilar (ABP 501), between 1 January 2003 and 31 December 2020. Inclusion criteria were: age > 18 years, starting and last ADA treatment date known. The reasons for ADA interruption were classified as primary inefficacy, secondary inefficacy, switch (from originator to ABP 501) and adverse events (e.g., infections, cancer onset, neurological or cardiological diseases, death, etc.).

For each patient, we collected demographic data, pharmacological therapy (i.e., csDMARDs or concomitant corticosteroid use at initiation of ADA treatment) and autoimmune profile (i.e., presence/absence of rheumatoid factor (RF) and anti-cyclic citrullinated peptide antibody (ACPA) in RA or human leukocyte antigen B27 (HLA-B27) positivity in axSpA). Enrolled patients were clustered in three groups: naive to ADA when receiving originator (oADA group), naive to ADA when receiving ABP 501 (bADA group), switcher from originator to ABP 501 (sADA group). The sADA group was composed of patients who had switched from originator to biosimilar in accordance with the indications of sustainability proposed by the Italian drugs agency (AIFA—Agenzia Italiana del Farmaco).

### 2.2. Statistical Analysis

Descriptive data were presented by medians (interquartile range) for continuous data or as numbers (percentages) for categorical data. Differences between groups were analysed with a two-tailed chi-square or Kruskal–Wallis test. The ADA retention rate was considered the interval of time in which the patients received the treatment regardless of dosing or administration changes. Subjects in remission were censored at the last date in which they received ADA. Retention rate curves were computed by the Kaplan–Meier method and compared statistically by a stratified log rank test.

A Cox regression analysed the effect on the retention rate of the following risk factors: age, sex, disease duration, diagnosis (RA, PsA or axSpA), ADA received (original, ABP 501, both naive and switched), and year of ADA prescription. Statistical significance was set at *p*-value < 0.05.

All analyses were performed using Medcalc statistical software, version 18.2.1 (MedCalc Software Ltd., Ostend, Belgium).

## 3. Results

### 3.1. Baseline Clinical Characteristics

The enrolled patients in the oADA, bADA and sADA numbered 724, 129 and 193, respectively; the total observation period was 9805.6 patient-months. The median periods of observation in the three groups of patients were, respectively, 37.7 (12.7–81.7), 10.3 (4–14.8) and 12.9 (8.3–15.8) months. The groups were different in terms of age (oADA patients were slightly younger), disease prevalence (RA and PsA were less and more common, respectively, in bADA than in the other two groups) and duration (sADA had the longest). For more details, see Table 1.

#### 3.1.1. RA Cohort

Four hundred and thirty RA patients were enrolled, 311 in the oADA group, 40 in the bADA group and 79 in the sADA group. The majority, 327 (76%), were females. The median age was 60.8 (51.5–69.1) years, whereas the median disease duration was 9.1 (3–16.5) years. Patients in the oADA group were younger, whereas the ones in the sADA group had a longer disease duration. The highest ACPA positivity prevalence was in bADA, while csDMARDs and corticosteroid use in sADA were less common than in bADA and oADA. For more details, see Table 2.

#### 3.1.2. PsA Cohort

Three hundred sixteen PsA patients were enrolled, 216 in the oADA group, 52 in the bADA group and 48 in the sADA group. The majority, 160 (50.6%), were females. The median age was 54.3 (45.1–62.9) years, whereas the median disease duration was 4.4 (1.6–10) years. Patients in the oADA group were younger, whereas the ones in the sADA group had a longer disease duration. For more details, see Table 3.

#### 3.1.3. AxSpA Cohort

Three hundred axSpA patients were enrolled, 197 in the oADA group, 37 in the bADA group and 66 in the sADA group. The majority, 171 (57%), were males. The median age was 49.3 (40.8–57.5) years, whereas the median disease duration was four (1.1–10) years. Patients in the bADA group were younger, whereas the ones in the sADA group had a longer disease duration. The HLA-B27 positivity prevalence was lowest in the bADA group. For more details, see Table 4.

### 3.2. Drug Survival

The 18-month retention rate of ADA in the overall cohort of patients was 82.6%. The 18-month retention rates of ADA in RA, PsA and axSpA patients were 81.9%, 81.1% and 85.2%, respectively, without statistically significant differences. The Kaplan–Meier curve in Figure 1 shows the different 18-month retention rates in the groups of patients according to oADA, bADA and sADA. sADA were more persistent than bADA and oADA (88.0%, 84.0% and 81.5%, respectively; *p* > 0.05).

### 3.3. Reasons for Discontinuation

Overall, 158 ADA treatments were interrupted. The reasons for treatment interruption were secondary inefficacy (76, 48.1%), primary inefficacy (60, 38%) and adverse events (22, 13.9%). Among the adverse events, the most frequent were cutaneous reactions (*n* = 8) followed by serious infections (*n* = 3), neoplasia (*n* = 3: melanoma (*n* = 1), uterine cancer (*n* = 1), prostate cancer (*n* = 1)), malaise (*n* = 3), injection site reaction (*n* = 2), alopecia (*n* = 1), autoimmune complication (systemic lupus erythematosus, *n* = 1) and neurologic complication (*n* = 1). In the oADA group, 45 patients interrupted the treatment due to primary inefficacy, 62 due to secondary inefficacy and 18 due to adverse events. In the bADA group, six patients interrupted the treatment due to primary inefficacy, six due to secondary inefficacy and one due to adverse events. In the sADA group, nine patients interrupted the treatment due to primary inefficacy, eight due to secondary inefficacy and three due to adverse events.

### 3.4. Predictors of ADA Discontinuation

A Cox regression was used to analyse predictors of ADA discontinuation in our cohort of patients. The Cox regression identified the following factors as influencing the retention rate: axSpA diagnosis (Hazard Ratio 0.70, 95% Confidence Intervals (CI) 0.49–0.99; *p* = 0.04), switch from oADA to ABP 501 (HR 0.53, 95% CI 0.31–0.90; *p* = 0.02) and year of prescription (HR 1.04, 95% CI 1.01–1.08; *p* = 0.04).

## 4. Discussion

To the best of our knowledge, this is the first study to evaluate the retention rate of ABP 501 in a large cohort of real-life patients with RA, PsA and axSpA, both naïve and ADA-experienced. Overall, we retrospectively analysed 1046 patients treated with ADA, 322 of whom were treated with ABP 501. The 18-month retention rates for oADA, bADA and sADA were 81.5%, 84% and 88%, respectively, although without a significant difference. The 18-month ADA drug survival is comparable to those reported previously in similar real-life studies [14,15,16]. Our results highlight the overall good drug persistence of ABP 501 in both inflammatory arthritides patients who are naïve and experienced to ADA. In particular, ABP 501 patients switching from originator to biosimilar showed a high retention rate that was comparable to the ones reported for other ADA biosimilars such as GP2017 and SB5 [17].

In our cohort of patients, as expected, sADA showed a longer disease duration and a higher age, since it included patients previously treated with the originator. Interestingly, the bADA group included relatively fewer patients with RA, and we hypothesise that this may be due to the greater availability of treatments for RA when compared to PsA and axSpA.

Among RA patients, those in the sADA group had a significantly lower frequency of combination treatment with csDMARDs. This is in accordance with previous observations that up to one-third of patients treated with bDMARDs experience dose reduction or discontinuation of csDMARDs during the first two years of treatment [18]. RA patients in the sADA group were less frequently treated with corticosteroids. This finding may be due to these patients having been on a stable regimen with ADA originator prior to the medication switch. We observed a relatively lower prevalence of HLA-B27 positivity in the bADA group. This might be related to the slightly higher frequency of female patients in this group [19].

The most frequent reason for ADA treatment discontinuation in the three groups was inefficacy, similar to what has been reported in previous real-life observational studies [20,21,22]. In particular, secondary inefficacy, i.e., loss of response after an initial clinical improvement, was the main cause of treatment discontinuation in our cohort of patients. On the other hand, adverse events leading to ADA treatment interruption were relatively less common. In particular, the transition from ADA originator to ABP 501 appeared to be safe and well-tolerated.

Finally, we analysed the factors associated with drug survival with a Cox regression. We found that patients with axSpA were less likely to discontinue treatment with ADA. This result is in line with reports from other large multicentric registries such as MonitorNet and Nor-DMARD [23,24]. Moreover, patients switching from originator ADA to ABP 501 showed a lower risk of treatment discontinuation. Observational data on both other ADA biosimilars such as GP2017 and SB5 [17,25] and other TNFi biosimilars [26,27,28,29,30] showed that switching from originator to biosimilar was tolerated generally well. The only other factor associated with a negative effect on drug survival in our cohort of patients was the year of ADA prescription. We observed that the discontinuation rates were higher in those patients who started treatment in later years when compared to the ones who started treatment in earlier years. This effect has also been reported for other TNFi [22,31], and we suggest that this result could be influenced by the increasing number of treatments available for the therapy of inflammatory arthritides and the adoption of treat-to-target strategies.

Our study has some limitations. The main limitation of our study lies in its observational retrospective design, leading to a possible selection bias stemming from including patients with different discontinuation risks. However, we tried to minimise that risk by analysing the possible role of other variables, beyond oADA, bADA and sADA treatment, on drug survival. Another limitation of this study is that due to its retrospective design, we do not have data related to disease activity levels and their trends in time. Similarly, medication side effects besides the ones leading to treatment discontinuation were not recorded. Furthermore, the relatively small sample size and unequal number of patients treated with ADA or ABP 501, especially in some subgroups of inflammatory arthritides patients, could be a cause of bias. Moreover, our study has a relatively short follow-up period. Finally, given the retrospective observational nature of our study, we do not present any data on ADA pharmacokinetics or levels of anti-drug antibodies, especially in the population switching from originator to biosimilar.

In conclusion, our large retrospective study highlights that patients treated with ADA originator or with its biosimilar ABP 501 have an overall similar drug survival and a favourable safety profile. The main factors associated with a better drug survival were axSpA diagnosis and the switch from ADA originator to ABP 501, whereas the year of prescription seemed to have a negative effect. Further studies, especially in patients treated for a longer duration with ABP 501, should be encouraged in order to confirm our results.

## Figures and Tables

**Figure 1 jpm-12-00335-f001:**
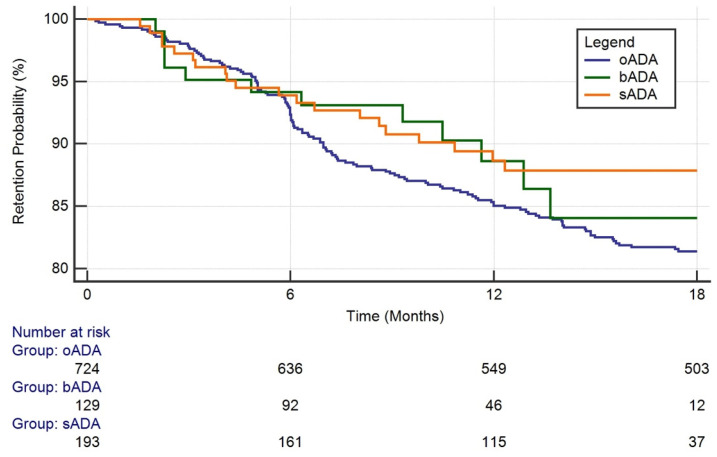
18-month retention rate according to oADA, bADA and sADA. oADA = naive to ADA when receiving the originator; bADA = naive to ADA when receiving ABP 501; sADA = switcher from originator to ABP 501.

**Table 1 jpm-12-00335-t001:** Cohort characteristics.

	oADA Group	bADA Group	sADA Group	*p* Value
*N*	724	129	193	-
M:F	300:424	42:87	88:105	>0.05 °
Age, median (IQR) (years)	53.9 (43.6–63.5)	56.0 (45.7–67.0)	57.7 (50.3–69.7)	<0.05 +
Disease duration, median (IQR) (years)	4.3 (1.4–11.0)	3.6 (1.4–9.8)	12.6 (7.5–19.6)	<0.05 +
Diagnosis, *n* (%) - RA - PsA - AxSpA	311 (43.0%) 216 (29.8%) 197 (27.2%)	40 (31.0%) 52 (40.3 %) 37 (28.7%)	79 (40.9%) 48 (24.9%) 66 (34.2%)	<0.05 °
Line, *n* (%) - 1 - 2 - 3 +	521 (71.9%) 146 (20.2%) 57 (7.9%)	89 (69.0%) 28 (21.7%) 12 (9.3%)	0 (0) 138 (71.5%) 55 (28.4%)	<0.05 °

Data are reported as median and interquartile range (IQR) and frequencies (number and %). RA = rheumatoid arthritis; PsA = psoriatic arthritis; AxSpA = axial spondylarthritis; oADA = naive to ADA when receiving the originator; bADA = naive to ADA when receiving ABP 501; sADA = switcher from originator to ABP 501. Line = line of biologic treatment. ° chi-square test; + Kruskal–Wallis test.

**Table 2 jpm-12-00335-t002:** RA cohort characteristics.

	oADA Group	bADA Group	sADA Group	*p* Value
*N*	311	40	79	-
M:F	77:234	5:35	21:58	>0.05 °
Age, median (IQR) (years)	59.3 (49.6–67.6)	59.9 (51.5–67.2)	66.5 (56.5–78.5)	<0.05 +
Disease duration, median (IQR) (years)	6.4 (2.4–15.5)	5.7 (2.4–11.9)	14.5 (9.4–23.8)	<0.05 +
RF positive, %	50.2%	50.0%	39.2%	>0.05 °
ACPA positive, %	54.7%	57.5%	40.5%	<0.05 °
Line, *n* (%) - 1 - 2 - 3 +	208 (66.9%) 75 (24.1%) 28 (9.0%)	26 (65.0%) 9 (22.5%) 5 (12.5%)	0 (0%) 50 (63.3%) 29 (36.7%)	<0.05 °
csDMARDs association, %	60.5%	72.5%	53.2%	<0.05 °
steroid association, %	54.7%	72.5%	39.2%	<0.05 °

Data are reported as median and interquartile range (IQR) and frequencies (number and %). oADA = naive to ADA when receiving the originator; bADA = naive to ADA when receiving ABP 501; sADA = switcher from originator to ABP 501. Line = line of biological treatment. ° chi-square test; + Kruskal–Wallis test.

**Table 3 jpm-12-00335-t003:** PsA cohort characteristics.

	oADA Group	bADA Group	sADA Group	*p* Value
*N*	216	52	48	-
M:F	108:108	18:34	30:18	<0.05 °
Age, median (IQR) (years)	53.4 (43.0–60.1)	61.3 (50.3–68.9)	55.7 (46.9–65.2)	<0.05 +
Disease duration, median (IQR) (years)	3.5 (1.2–8.0)	4.5 (1.2–7.6)	12.6 (5.8–19.1)	<0.05 +
Line, *n* (%) - 1 - 2 - 3 +	163 (75.5%) 40 (18.5%) 13 (6.0%)	34 (65.4%) 12 (23.1%) 6 (11.5%)	0 (0%) 38 (79.2%) 10 (20.2%)	<0.05 °
csDMARDs association, %	35.2%	42.3%	41.7%	>0.05 °

Data are reported as median and interquartile range (IQR) and frequencies (number and %). oADA = naive to ADA when receiving the originator; bADA = naive to ADA when receiving ABP 501; sADA = switcher from originator to ABP 501. Line = line of biological treatment. ° chi-square test; + Kruskal–Wallis test.

**Table 4 jpm-12-00335-t004:** AxSpA cohort characteristics.

	oADA Group	bADA Group	sADA Group	*p* Value
*N*	197	37	66	-
M:F	115:82	19:18	37:29	>0.05 °
Age, median (IQR) (years)	48.4 (39.7–56)	46.0 (41.1–56.1)	53.2 (46.1–61.6)	<0.05 +
Disease duration, median (IQR) (years)	2.2 (0.4–8.1)	3.0 (1.1–8.1)	10.3 (6.5–13.6)	<0.05 +
HLAB27 positive, %	73.1%	54.0%	74.2%	>0.05 °
Line, *n* (%) - 1 - 2 - 3 +	150 (76.1%) 31 (15.8%) 16 (8.1%)	29 (78.4%) 7 (18.9%) 1 (2.7%)	0 (0%) 50 (75.8%) 16 (24.2%)	<0.05 °

Data are reported as median and interquartile range (IQR) and frequencies (number and %). oADA = naive to ADA when receiving the originator; bADA = naive to ADA when receiving ABP 501; sADA = switcher from originator to ABP 501. Line = line of biological treatment. ° chi-square test; + Kruskal–Wallis test.

## Data Availability

Data can be made available from the corresponding author upon request.

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
