# Peer review of "Adalimumab and ABP 501 in the Treatment of a Large Cohort of Patients with Inflammatory Arthritis: A Real Life Retrospective Analysis"

_jpm, 2022, doi:10.3390/jpm12030335_

Round 1

Reviewer 1 Report

This study is a nicely done multicenter, observational retrospective study aiming to evaluate the retention rate of adalimumab biosimilar ABP 501 in patients with inflammatory arthritis naive to ADA original or its ABP 501 biosimilar and in those who switched from ADA original to ABP 501 biosimilar.

This study is of importance given lack of real-world data on the efficacy and safety of novel biosimilar agents currently being used in rheumatology practice relative to the original formulations as measured by drug retention rates.

I have a few questions regarding the methodology:

-It would be nice to note something about whether the registry used in the study was ever evaluated for the accuracy of drug dispensation or diagnoses that were evaluated in the study (even if done in previous studies and not by the authors of this paper, and then please cite these previous studies in your references).

-Was only a single drug dispensation date used for inclusion in the study or two or more consecutive drug dispensations to increase the accuracy of the data?  This is done in some of these sort of studies to increase the likelihood that the patient was taking the medication as prescribed.  Also along this line,  were any grace periods for medication refilling taken into account when calculating drug persistence?  Some studies allow for the passage of a few months between medication refilling as a reasonable grace period for medication refilling before signifying that there is lack of drug persistence [see for example, Haddad A. et al. "Treatment persistence of biologics among patients with psoriatic arthritis." Arthritis Res Ther. 2021 Jan 29;23(1):44].

-Was there any data available on disease activity level or medication side effects in the registry?  If not, as is commonly the case with registry data, this should be listed as a limitation in the study, as access to this data would be important in assessing lack of efficacy/safety of the medication being studied and drug persistence used in this study is really being used as a surrogate for drug efficacy/safety as correctly asserted by the authors in the paper but is not as accurate as using original data on disease activity level or side effects causing drug discontinuation.

-Please highlight what was the reason for medication change from ADA originator to ADA biosimilar?  Were all of these patients previously stable on ADA originator and were switched due to commercial health insurance or national insurance guidelines (or patient request? other reasons?)

-No information was given on the institutional review board (Helsinki) approval for this study in the methods section as is commonly done in many journals (I saw it is listed at the end of the paper).  If this is required by this journal, please also add it in the methods section.

-In lines 156-157 of the results section, which types of neoplasia and what autoimmune complication caused discontinuation of treatment?

-Table 2-when discussing % of patients on corticosteroids, can you please give average doses used if you have access to this information?

In addition, I have a few grammatical errors with suggested changes:

Line 38-39: I would change "In each group the majority of them had a diagnosis of rheumatoid arthritis" to "In each group, the majority of patients had..."

Line 45: I would change "even not statistically significant" to "even though not statistically significant"

Line 57:  I would change "In the last years" to "in recent years"

Line 60: I would change "has been approved" to "was approved"

Line 61: I would change `'demonstrated to be" to "was demonstrated to be"

Line 70:  I would add a period after "real-world clinical practice" instead of the apostrophe.

Line 96:  I would change "csDMARDs or steroids associated at ADA at the beginning of treatment" to "csDMARDs or concomitant corticosteroid use at initiation of ADA treatment"

Line 97: Please change "anti–citrullinated protein antibody [ACPA]" to "anti-cyclic citrullinated peptide antibody [ACPA]"

Line 98: Please change "human lymphocyte antigen B27 [HLA-B27] positivity" to human leukocyte antigen B27 [HLA-B27] positivity"

Line 107-108: I would change "received the treatment regardless changes of dose or administration frequency" to "received the treatment regardless of dosing or administration changes"

Line 111:  I would change "of the following risk factor" to the plural "of the following risk factors"

Line 117: I would  change "The enrolled patients in the oADA, bADA and sADA were respectively 724, 129 and 193" to "The enrolled patients in the oADA, bADA and sADA were 724, 129 and 193, respectively"

Line 121: I would change "RA and PsA were respectively less and more common in bADA than in the other two groups" to "RA and PA were less and more common, respectively, in bADA than in the other two groups"

Line 130:  I would change "For a better comparison of groups characteristics" to "For a better comparison of group characteristics" (group should be written in the singular)

Line 132:  I would change "in whole cohort" to "in the entire cohort"

Line 133: I would change "csDMARDs and steroid association in sADA" to "csDMARs and corticosteroid-use in sADA..."

Line 135:  Please change "For more details, see Table 2-4" to "For more details, see Tables 2-4" (Tables in the plural form)

Line 152:  Please change "(respectively 88.0%, 84.0%, and 81,5%; p > 0.05)" to "(88.0%, 84.0% and 81.5%; respectively)"

Line 153:  Please type out the number 158 at the beginning of a sentence or write something like "Overall, 158 ADA treatments were interrupted" so that you do not have to type out "One-hundred and fifty-eight" as sentences are not typically started with a number not written out.

Line 153:  I suggest changing "The reasons of treatment interruption" to "The reasons for treatment interruption"

Line 178:  I would change "Our results highlighted" to "Our results highlight" in the present tense

Line 180:  Please change "an high retention rate" to "a high retention rate"

Line 181:  I would change "comparable to the ones reported for other ADA biosimilar such as GP2017 and SB5" to "comparable to the ones reported for other ADA biosimilars such as GP2017 and SB5" with biosimilars being in the plural form

Line 185:  Please change "the bADA group included a relatively fewer patients with RA" to "the bADA group included relatively fewer patients with RA"

Lines 187-189:  I suggest changing "Among the RA patients sADA showed a significantly lower frequency of combination treatment with csDMARDs, this could be due to the fact that it has been observed up to one third of the patients treated with bDMARDs reduce or interrupt the treatment of csDMARDs in the first two years of treatment" to something along the lines of "Among RA patients, those in the sADA group had a significantly lower frequency of combination treatments with csDMARDs.  This is in accordance with previous observations that up to one-third of patients treated with bDMARDs experience dose reduction or discontinuation of csDMARDs during the first two years of treatment."

Lines 190-192:  I suggest changing "RA patients in sADA group were less frequently treated with corticosteroids, it is unsurprisingly since this group is constituted by patients in previous stable treatment with ADA originator" to something along the lines of "RA patients in the sADA group were less frequently treated with corticosteroids. This finding my be due to these patients having been on a stable regimen with ADA originator prior to the medication switch."  This would eliminate a run-on sentence.

Lines 192-194: I would change "We observed a relatively lower prevalence of HLA-B27 positivity in the bADA group, this could be related to the  slightly higher frequence of female patients in this group" to "We observed a relatively lower prevalence of HLA-B27 positivity in the bADA group.  This might be related to the slightly higher frequency of female patients in this group..."  This would eliminate a run-on sentence.

Line 195:  I would change "The most frequent reason of ADA treatment discontinuation" to "The most frequent reason for ADA treatment discontinuation"

Line 204: I would change "This result is in line with what reported by other large multicentric registries" to "This result is in line with reports from other large multicentric registries"

Line 216-218:  I wold change "The main limitation is its observational retrospective design, therefore patients with a possible different discontinuation risk, producing a selection bias, could have led to incorrect analysis results" to something along the lines of  "The main limitation of our study lies in its observational retrospective design, leading to possible selection bias stemming from including patients with different discontinuation risks in our study" --this wording would eliminate the run-on sentence and clarify the content

Lines 226-231:  I would make a few changes in the concluding statements, as follows: "In conclusion,  our large retrospective study highlights that patients treated with ADA originator or with its biosimilar ABP 501 have overall similar drug survival and a favorable safety profile. The main factors associated with better drug survival were axSpA diagnosis and switch from ADA originator to ABP 501, whereas the year of prescription seems to have a negative effect. Further studies, especially in patients treated for a longer duration with ABP 501, should be advocated to confirm our results."

Author Response

This study is a nicely done multicenter, observational retrospective study aiming to evaluate the retention rate of adalimumab biosimilar ABP 501 in patients with inflammatory arthritis naive to ADA original or its ABP 501 biosimilar and in those who switched from ADA original to ABP 501 biosimilar.

This study is of importance given lack of real-world data on the efficacy and safety of novel biosimilar agents currently being used in rheumatology practice relative to the original formulations as measured by drug retention rates.

I have a few questions regarding the methodology:

-It would be nice to note something about whether the registry used in the study was ever evaluated for the accuracy of drug dispensation or diagnoses that were evaluated in the study (even if done in previous studies and not by the authors of this paper, and then please cite these previous studies in your references).

  • The study did not evaluated the accuracy of drug dispensation or diagnoses. In Italy, the dispensing of biological drugs in rheumatology is linked to the preparation of a therapeutic plan which must be drawn up every three to six months in the face of a clinical evaluation. Every therapeutic plan must be linked with a specific clinical diagnosis (in our study RA, PsA or axSpA).

-Was only a single drug dispensation date used for inclusion in the study or two or more consecutive drug dispensations to increase the accuracy of the data?  This is done in some of these sort of studies to increase the likelihood that the patient was taking the medication as prescribed.  Also along this line,  were any grace periods for medication refilling taken into account when calculating drug persistence?  Some studies allow for the passage of a few months between medication refilling as a reasonable grace period for medication refilling before signifying that there is lack of drug persistence [see for example, Haddad A. et al. "Treatment persistence of biologics among patients with psoriatic arthritis." Arthritis Res Ther. 2021 Jan 29;23(1):44]. 

  • Each patient that received at least one dose of adalimumab was included in the study. Among 1046 treated patients, only 33 were registered as treated for less than 1 months. Among these 33 patients: four withdrew the treatment due to adverse events (n=3) or inefficacy (worsening, n=1); six were lost on follow-up; twenty-three continued treatment but were censored given the end of the observation period (December 2020) for the study. Each patient was regularly followed up by their referral center every three to six months. Every clinician reported the discontinuation date, and the relative discontinuation reason, according to the last administered dose of adalimumab. Transient drug withdrawal (eg: for concomitant infection or surgery) were not reported.

-Was there any data available on disease activity level or medication side effects in the registry?  If not, as is commonly the case with registry data, this should be listed as a limitation in the study, as access to this data would be important in assessing lack of efficacy/safety of the medication being studied and drug persistence used in this study is really being used as a surrogate for drug efficacy/safety as correctly asserted by the authors in the paper but is not as accurate as using original data on disease activity level or side effects causing drug discontinuation.

  • Due to retrospective nature of this study unfortunately we do not have a complete data set regarding disease activity level or medication side effects not leading to treatment discontinuation. These limitations have been listed in the discussion section.

-Please highlight what was the reason for medication change from ADA originator to ADA biosimilar?  Were all of these patients previously stable on ADA originator and were switched due to commercial health insurance or national insurance guidelines (or patient request? other reasons?)

  • The sADA group was composed of patients who had switched from originator to biosimilar in accordance with the indications of sustainability proposed by the Italian drugs agency (AIFA-Agenzia Italiana del Farmaco). This sentence has been added in the methods section.

-No information was given on the institutional review board (Helsinki) approval for this study in the methods section as is commonly done in many journals (I saw it is listed at the end of the paper).  If this is required by this journal, please also add it in the methods section.

  • It has been added in the methods section.

-In lines 156-157 of the results section, which types of neoplasia and what autoimmune complication caused discontinuation of treatment?

  • The types of neoplasia and autoimmune complication have been reported in the results section.

-Table 2-when discussing % of patients on corticosteroids, can you please give average doses used if you have access to this information?

  • Unfortunately, we do not have access to this information

In addition, I have a few grammatical errors with suggested changes:

  • Thank you for the kind suggestions.

he majority of them had a diagnosis of rheumatoid arthritis" to "In each group, the majority of patients had..."

Line 45: I would change "even not statistically significant" to "even though not statistically significant"

Line 57:  I would change "In the last years" to "in recent years"

Line 60: I would change "has been approved" to "was approved"

Line 61: I would change `'demonstrated to be" to "was demonstrated to be"

Line 70:  I would add a period after "real-world clinical practice" instead of the apostrophe.

Line 96:  I would change "csDMARDs or steroids associated at ADA at the beginning of treatment" to "csDMARDs or concomitant corticosteroid use at initiation of ADA treatment"

Line 97: Please change "anti–citrullinated protein antibody [ACPA]" to "anti-cyclic citrullinated peptide antibody [ACPA]"

Line 98: Please change "human lymphocyte antigen B27 [HLA-B27] positivity" to human leukocyte antigen B27 [HLA-B27] positivity"

Line 107-108: I would change "received the treatment regardless changes of dose or administration frequency" to "received the treatment regardless of dosing or administration changes"

Line 111:  I would change "of the following risk factor" to the plural "of the following risk factors"

Line 117: I would  change "The enrolled patients in the oADA, bADA and sADA were respectively 724, 129 and 193" to "The enrolled patients in the oADA, bADA and sADA were 724, 129 and 193, respectively"

Line 121: I would change "RA and PsA were respectively less and more common in bADA than in the other two groups" to "RA and PA were less and more common, respectively, in bADA than in the other two groups"

Line 130:  I would change "For a better comparison of groups characteristics" to "For a better comparison of group characteristics" (group should be written in the singular)

Line 132:  I would change "in whole cohort" to "in the entire cohort"

Line 133: I would change "csDMARDs and steroid association in sADA" to "csDMARs and corticosteroid-use in sADA..."

Line 135:  Please change "For more details, see Table 2-4" to "For more details, see Tables 2-4" (Tables in the plural form)

Line 152:  Please change "(respectively 88.0%, 84.0%, and 81,5%; p > 0.05)" to "(88.0%, 84.0% and 81.5%; respectively)"

Line 153:  Please type out the number 158 at the beginning of a sentence or write something like "Overall, 158 ADA treatments were interrupted" so that you do not have to type out "One-hundred and fifty-eight" as sentences are not typically started with a number not written out.

Line 153:  I suggest changing "The reasons of treatment interruption" to "The reasons fortreatment interruption"

Line 178:  I would change "Our results highlighted" to "Our results highlight" in the present tense

Line 180:  Please change "an high retention rate" to "a high retention rate"

Line 181:  I would change "comparable to the ones reported for other ADA biosimilar such as GP2017 and SB5" to "comparable to the ones reported for other ADA biosimilars such as GP2017 and SB5" with biosimilars being in the plural form

Line 185:  Please change "the bADA group included a relatively fewer patients with RA" to "the bADA group included relatively fewer patients with RA"

Lines 187-189:  I suggest changing "Among the RA patients sADA showed a significantly lower frequency of combination treatment with csDMARDs, this could be due to the fact that it has been observed up to one third of the patients treated with bDMARDs reduce or interrupt the treatment of csDMARDs in the first two years of treatment" to something along the lines of "Among RA patients, those in the sADA group had a significantly lower frequency of combination treatments with csDMARDs.  This is in accordance with previous observations that up to one-third of patients treated with bDMARDs experience dose reduction or discontinuation of csDMARDs during the first two years of treatment."

Lines 190-192:  I suggest changing "RA patients in sADA group were less frequently treated with corticosteroids, it is unsurprisingly since this group is constituted by patients in previous stable treatment with ADA originator" to something along the lines of "RA patients in the sADA group were less frequently treated with corticosteroids. This finding my be due to these patients having been on a stable regimen with ADA originator prior to the medication switch."  This would eliminate a run-on sentence.

Lines 192-194: I would change "We observed a relatively lower prevalence of HLA-B27 positivity in the bADA group, this could be related to the  slightly higher frequence of female patients in this group" to "We observed a relatively lower prevalence of HLA-B27 positivity in the bADA group.  This might be related to the slightly higher frequency of female patients in this group..."  This would eliminate a run-on sentence.

Line 195:  I would change "The most frequent reason of ADA treatment discontinuation" to "The most frequent reason for ADA treatment discontinuation"

Line 204: I would change "This result is in line with what reported by other large multicentric registries" to "This result is in line with reports from other large multicentric registries"

Line 216-218:  I wold change "The main limitation is its observational retrospective design, therefore patients with a possible different discontinuation risk, producing a selection bias, could have led to incorrect analysis results" to something along the lines of  "The main limitation of our study lies in its observational retrospective design, leading to possible selection bias stemming from including patients with different discontinuation risks in our study" --this wording would eliminate the run-on sentence and clarify the content

Lines 226-231:  I would make a few changes in the concluding statements, as follows: "In conclusion,  our large retrospective study highlights that patients treated with ADA originator or with its biosimilar ABP 501 have overall similar drug survival and a favorable safety profile. The main factors associated with better drug survival were axSpA diagnosis and switch from ADA originator to ABP 501, whereas the year of prescription seems to have a negative effect. Further studies, especially in patients treated for a longer duration with ABP 501, should be advocated to confirm our results."

Reviewer 2 Report

Present manuscript by Becciolini et al. attempts to evaluate the retention rate of adalimumab and its biosimilar in a reasonably big group of patients. Manuscript is focusing on important clinical question and largely the results are described sufficiently. I suggest minor revision with changes in some sections and workflow.

Line 79:

Though the manuscript says investigation of retention rate in three types of arthritis, the comparisons made are more with regard to differences in three therapy groups and not disease subtypes precisely. Authors should also make within group comparisons for the parameters where they are lacking.

Line 115:

I would suggest to make results divided into sections based on disease or divided on the basis of choice of therapy, so that it is easy to follow and better if some data and differences can be shown as figures also. As of now it more looks like a case series. It will be benefit a lot if some descriptive titles for results sections can be considered.

Line 122: Authors should also give separate disease specific disease duration values in the analysis, to see whether that was comparable between groups. The p values in the last column appear to be between group significance. Authors should also consider making within group analysis and mention in the descriptive results.

Table 2, Table 3 and Table 4 can then be included under separate headings and description of within disease comparisons.

Author Response

Authors response to reviewer comments:

Line 79:

Though the manuscript says investigation of retention rate in three types of arthritis, the comparisons made are more with regard to differences in three therapy groups and not disease subtypes precisely. Authors should also make within group comparisons for the parameters where they are lacking.

  • Data regarding the three types of arthritis have been added.

Line 115:

I would suggest to make results divided into sections based on disease or divided on the basis of choice of therapy, so that it is easy to follow and better if some data and differences can be shown as figures also. As of now it more looks like a case series. It will be benefit a lot if some descriptive titles for results sections can be considered.

  • The results section has been modified as suggested.

Line 122: Authors should also give separate disease specific disease duration values in the analysis, to see whether that was comparable between groups. The p values in the last column appear to be between group significance. Authors should also consider making within group analysis and mention in the descriptive results.

  • Data regarding the single diseases have been reported as suggested in the results section.

Table 2, Table 3 and Table 4 can then be included under separate headings and description of within disease comparisons.

  • The tables have been reported under separate headings as suggested.